# Assessing Ethnic Inequalities in Diagnostic Interval of Common Cancers: A Population-Based UK Cohort Study

**DOI:** 10.3390/cancers14133085

**Published:** 2022-06-23

**Authors:** Tanimola Martins, Gary Abel, Obioha C. Ukoumunne, Sarah Price, Georgios Lyratzopoulos, Frank Chinegwundoh, William Hamilton

**Affiliations:** 1College House St Luke’s Campus, College of Medicine and Health, University of Exeter, Magdalen Road, Exeter EX1 2LU, UK; s.j.price@exeter.ac.uk (S.P.); w.hamilton@exeter.ac.uk (W.H.); 2National Institute for Health and Care Research (NIHR), Applied Research Collaboration (ARC) South West Peninsula (PenARC), University of Exeter, Exeter EX1 2LU, UK; g.a.abel@exeter.ac.uk (G.A.); o.c.ukoumunne@exeter.ac.uk (O.C.U.); 3Epidemiology of Cancer Healthcare & Outcomes (ECHO) Group, University College London, 1–19 Torrington Place, London WC1E 7HB, UK; y.lyratzopoulos@ucl.ac.uk; 4Barts Health NHS Trust & Department of Health Sciences, University of London, London WC1E 7HB, UK; frank.chinegwundoh@nhs.net

**Keywords:** primary care, diagnostic interval, ethnic inequalities, early detection, diagnostic pathway, symptomatic cancer

## Abstract

**Simple Summary:**

UK ethnic minorities have poorer outcomes of some cancers and are less likely to report positive health care experiences. We wanted to determine whether these differences are related to inequalities in time to diagnosis once patients sought medical help with cancer symptoms. We found that in five of the seven cancers studied, the minority groups experienced a longer time to diagnosis when compared with the White group. However, the differences were small and unlikely to be the sole explanation for the ethnic variation in cancer outcomes. Nonetheless, addressing such differences will help to improve trust and care experiences among ethnic minority groups.

**Abstract:**

Background: This study investigated ethnic differences in diagnostic interval (DI)—the period between initial primary care presentation and diagnosis. Methods: We analysed the primary care-linked data of patients who reported features of seven cancers (breast, lung, prostate, colorectal, oesophagogastric, myeloma, and ovarian) one year before diagnosis. Accelerated failure time (AFT) models investigated the association between DI and ethnicity, adjusting for age, sex, deprivation, and morbidity. Results: Of 126,627 eligible participants, 92.1% were White, 1.99% Black, 1.71% Asian, 1.83% Mixed, and 2.36% were of Other ethnic backgrounds. Considering all cancer sites combined, the median (interquartile range) DI was 55 (20–175) days, longest in lung [127, (42–265) days], and shortest in breast cancer [13 (13, 8–18) days]. DI for the Black and Asian groups was 10% (AFT ratio, 95%CI 1.10, 1.05–1.14) and 16% (1.16, 1.10–1.22), respectively, longer than for the White group. Site-specific analyses revealed evidence of longer DI in Asian and Black patients with prostate, colorectal, and oesophagogastric cancer, plus Black patients with breast cancer and myeloma, and the Mixed group with lung cancer compared with White patients. DI was shorter for the Other group with lung, prostate, myeloma, and oesophagogastric cancer than the White group. Conclusion: We found limited and inconsistent evidence of ethnic differences in DI among patients who reported cancer features in primary care before diagnosis. Our findings suggest that inequalities in diagnostic intervals, where present, are unlikely to be the sole explanation for ethnic variations in cancer outcomes.

## 1. Background

Robust research evidence on the causes of ethnic inequalities in cancer outcomes in the UK is needed to design effective solutions. UK ethnic minorities have poorer outcomes for some cancers compared to the British White majority. For instance, Asian and Black women with breast cancer, and Black men with prostate cancer have poorer survival rates than their British White counterparts [1,2,3]. Additionally, patients from these groups are more likely than the White group to report suboptimal healthcare experiences in both primary and specialist care [4,5,6,7].

The causes of ethnic inequalities in cancer outcomes are complex and poorly understood, particularly in the context of a universally accessible healthcare system. Asian and Black groups in the UK have poorer awareness of cancer symptoms, are less likely to take up screening [8,9,10,11], and may delay help seeking when symptomatic [12,13]. Among those presenting symptomatically in primary care, these major ethnic groups are also less likely to fully disclose symptoms during initial consultations [14], and have more pre-referral consultations in primary care [15,16]. Furthermore, a recent study of patients presenting with lower urinary tract symptoms in primary care suggests that the general practitioner (GP) may be hesitant to investigate Asian and Black patients for possible prostate cancer [14]. Our vignette-based study showed that when presented with hypothetical scenarios (including risks, symptoms, and likely investigations), Black men were less willing to opt for primary care investigation for prostate cancer, particularly when the perceived risk of cancer was low [17].

These factors may lead to prolonged diagnostic intervals—the period between the first symptomatic consultation in primary care and definitive diagnosis—and poorer cancer outcomes. There is limited UK evidence of ethnic variation in diagnostic intervals [13]. UK Asian and Black patients with breast cancer were more likely to experience prolonged diagnostic intervals than the White group, although all studies had significant methodological limitations [12,13,18].

In the present study, we used linked data from primary care, secondary care, and the national cancer registry on patients diagnosed with cancer between 2006 and 2016, focusing on ethnic differences.

## 2. Methods

### 2.1. Study Design and Data Sources

This population-based cohort study of English patients diagnosed with one of seven common cancers used data from the Clinical Practice Research Datalink (CPRD-Aurum) with linkage to Hospital Episode Statistics (HES), and the National Cancer Registration and Analysis Service (NCRAS) cancer registry data. The scope and strengths of CPRD-linked data are well documented [19,20,21,22,23].

### 2.2. Participants

Eligible participants were aged at least 40 years on their date of diagnosis (index date), with an incident cancer recorded in the cancer registry between 1 January 2006 and 31 December 2016. We excluded patients diagnosed with a cancer atypical for their sex (e.g., male breast/cervix), and those diagnosed via screening. Additionally, we excluded patients with no primary care attendance, or with no cancer-specific features recorded in the year before diagnosis, and those with missing ethnicity records in the CPRD and HES (see below).

### 2.3. Study Variables

*Cancer sites:* Using the NCRA data, we extracted patient records on the four most common cancers [lung (ICD10 C34), breast (C50), prostate (C61), colorectal (C18–C20)], three cancers commonly diagnosed in ethnic minority groups [oesophagus (C15), stomach (C16)), myeloma (C90)], and ovarian cancer (C56), for which we have sufficient data for analysis. (1) We merged the oesophagus and stomach into the oesophagogastric because they share diagnostic features and suspected-cancer referral criteria.

*Ethnicity:* Patients’ ethnicity was identified from CPRD codes, or HES data if missing in the CPRD, as recommended [22,24]. The processes involved in ethnicity data extraction are detailed elsewhere [25]. Briefly, we extracted and collapsed all ethnicity records from the CPRD into five major ethnic categories (White, Asian, Black, Mixed, and Other), in line with the 2001 UK census groupings. For individuals with multiple ethnicity codes, we adapted Mathur et al.’s algorithm to assign a single best ethnicity based on the most frequently and most recently recorded codes [26]. Those with missing ethnicities in both databases were excluded from the analyses. However, we encountered a non-specific ethnicity code in the CPRD labelled “British or British Mixed”, which may describe someone of British-Black, British-Asian, British-White, Mixed Black, Mixed Asian, Mixed White, or Other Mixed ethnicity. Patients with this ethnicity label had no substitute code in the CPRD, but 96% were recorded as White in HES. Therefore, we replaced this group with HES ethnicity records rather than assigning them to a particular group.

*Features of possible cancer:* We identified features of possible cancer (Table 1) using codes in the CPRD [27] based on the symptoms, signs, or blood test results in the original or revised National Institute for Health and Care Excellence (NICE) guidelines [28,29]. The index feature was the first feature recorded in the year before diagnosis [30]. Features recorded more than one year before diagnosis are less likely to be caused by cancer [31].

*Milestone dates and diagnostic interval:* The diagnostic interval (DI) was the time from the date of the index feature to the date of cancer diagnosis [32]. The date of cancer diagnosis was defined as the earliest date recorded in the NCRAS data.

*Other variables:* Patients’ age, sex, and multi-morbidities were identified from the CPRD. For age, we assigned everyone a nominal birthday of 1st July, as only birth year data are available. Socioeconomic deprivation was measured using quintiles of the 2015 Index of Multiple Deprivation (IMD) available via the CPRD linkage [23]. Data on morbidities—recorded before cancer diagnosis—were extracted from the CPRD using medical codes relating to 36 long-term conditions described elsewhere [33]. A patient-level morbidity score was derived as the sum of the General-outcome weighting assigned to each of their conditions, as previously described [34]. Patients with none of these conditions were assigned a score of zero. The score was entered into analysis as quartile-based groups of increasing morbidity burden.

### 2.4. Statistical Analyses

Accelerated Failure Time (AFT) models examined associations between diagnostic intervals and ethnicity. These models were chosen over conventional Cox Proportional Hazard Models because the coefficients from the AFT models (time ratios) are readily interpretable, where a time ratio > 1 indicates a longer DI, and a time ratio < 1 indicates shorter DI in the Black, Asian, Mixed, or Other groups compared to the White group. Analyses were run using Weibull, log-logistic, generalised gamma, and exponential distributions, with the Akaike Information Criterion (AIC) used to select the best parameterisation of the data. We reported crude and adjusted time ratios. Multivariable analyses adjusted for age, sex, IMD, and morbidity scores, and used information sandwich (“robust”) standard errors to allow for lack of independence of observation within practices.

However, DI varies significantly by cancer type, age, sex, deprivation, and the presence of comorbidities, all of which are also associated with ethnicity [35,36,37,38,39]. Therefore, to determine the best-fit model for the association between DI and ethnicity, we fitted several AFT models, including interactions between cancer and ethnicity, age, sex, IMD, and morbidity score. Each of the five interaction terms was tested in turn with a single joint test, and statistically significant interaction terms (*p* < 0.05) were retained in a final model including all interactions. Alongside age, IMD, and morbidity score, we found strong evidence of interaction between cancer type and ethnicity (*p* < 0.001), suggesting that the association between ethnicity and DI varies by cancer type. To illustrate these differences, we present analyses of DI with ethnicity stratified by cancer. The regression results are reported as time ratios, and differences in diagnostic intervals between the ethnic groups are reported as average marginal effects. All analyses were carried out in Stata v16.1 (StataCorp, College Station, TX, USA), and the reporting was guided by the STROBE framework for reporting observational studies [40].

## 3. Results

### 3.1. Participant Characteristics

Our cohort contained 220,702 potential participants with one of the seven cancers. Of these, 94,075 (42.6%) were excluded as follows: not having a recorded cancer feature in the year before diagnosis (n = 71,750), screen-detected [breast (n = 16,366) and colorectal (3341)], diagnosed via death certificates only (n = 743) or with a cancer atypical for their sex (n = 340), aged < 40 years (n = 477), and those with missing record on ethnicity (n = 1055) and diagnostic route (n = 3). The process of exclusion and the overall proportion of cancer-specific features are illustrated in Appendix A, respectively. Table 2 shows the demographics of the 126,627 patients included in the study. Overall, 116,640 (92.1%) were of White ethnic background, 2522 (1.99%) Black, 2159 (1.71%) Asian, 2321 (1.83%) Mixed, and 2985 (2.36%) Other ethnicity. Around three-fifths of the participants were males, with the proportion of males ranging from 54% in the Asian to 70% in the Black group. At diagnosis, Asian and Black patients were younger and lived in the most deprived areas compared to patients from other ethnic groups. The proportion with co-morbidity was slightly higher in the Black (94.3%) and Mixed groups (95.1%) but lower in the Other group (88.3%) compared with the White group (93.3%).

### 3.2. Index Features of Cancer

Breast cancer was unique in being dominated by a single presenting index feature (lump, 92.6%), while the remaining sites featured multiple non-specific features (Appendix A). The distribution of index features by site was broadly similar by ethnicity, although the proportions varied slightly for some sites (Appendix A).

### 3.3. Diagnostic Interval

Across the entire cohort, the median (interquartile range (IQR)) DI was 55 (20–175) days (Figure 1). The longest interval was observed in the lungs (median: 127, IQR: 42–265 days) and the shortest in breast cancer (median: 13, IQR: 13, 8–18 days). There were significant differences in DI by ethnicity, as described below.

### 3.4. Ethnic Differences in Diagnostic Interval

Across all sites, the median (IQR) DI was 55 (20–175) days in the White group, 61 (25–179) days in Black, 60 (17–176) days in Asian, 53 (18–182) days in the Mixed, and 45 (17–148) days in the Other ethnic group (Figure 1). After adjusting for age, sex, IMD, and morbidity score, DI for the Black and Asian groups was 10% (adjusted AFT ratio, 95%CI: 1.10, 1.05–1.14) and 16% (1.16, 1.10–1.22) longer, and 12% (0.88, 0.84–0.91) shorter in the Other group than for the White group. DI was similar between the Mixed and the White group (Table 3). There was strong evidence of interaction between site and age, IMD, and morbidity score (*p* < 0.001 for each of these), and between cancer site and sex (*p* = 0.04). The effect of sex was such that women were diagnosed somewhat faster than men in cancers affecting both sexes, with the largest difference for oesophagogastric cancer (AFT ratio = 1.17) and the smallest difference for lung and colorectal cancer (AFT ratio = 1.10 for both). Importantly, we found evidence of interaction between site and ethnicity (*p* < 0.001), suggesting that the effect of ethnicity on DI differs by cancer site. Table 3 shows the main findings from the analyses of the association between DI and ethnicity stratified by cancer. Specifically, DI was longer in the Asian and Black groups, respectively, with prostate [(adjusted AFT ratio, 95%CI 1.17, 1.09–1.27) and (1.09, 1.03–1.15)], colorectal [(1.37, 1.24–1.51) and (1.22, 1.09–1.35)], and oesophagogastric cancer [(1.33, 1.14–1.55) and (1.21, 1.05−1.41], alongside the Black group with myeloma (1.16, 1.00−1.35) and breast cancer (1.12, 1.04–1.21), than in the White group (Table 3). Conversely, DI for the Other group compared with the White group was shorter in lung (0.90, 0.85–0.95), prostate (0.87, 0.80–0.95), oesophagogastric cancer (0.84, 0.74–0.97), and myeloma (0.83, 0.66–1.03). DI was slightly longer for the Mixed group with lung cancer (1.06, 0.99–1.12) but similar in other sites relative to the White group. There was no evidence of a difference in DI between the Asian, Black, and White groups with lung and ovarian cancer alongside the Other group with breast, colorectal, and ovarian cancer.

## 4. Discussion

On average, across all cancers combined, DI was seven days longer in the Black group, eleven days longer in the Asian group, and nine days shorter in the Other ethnic group, all when compared with the White group. Site-specific estimates showed that DI in the Asian group with prostate, oesophagogastric, or colorectal cancer was, on average, 13, 20, and 24 days longer, respectively, compared with the White group. The corresponding figures for the Black group with myeloma, colorectal, oesophagogastric, prostate, or breast cancer were 15, 14, 13, 7, and 2 days longer than in the White group. Conversely, the Other group with lung, prostate, oesophagogastric cancer, or myeloma had shorter DI relative to the White group: about 12, 10, 10, and 16 days shorter, respectively. DI in the Mixed group with lung cancer was 7 days longer but similar across other sites relative to the White group.

### 4.1. Strengths and Limitations

The study was large and examined seven common cancers. The CPRD is the largest primary care database worldwide and is recognised for its high-quality data [19,20,21,22,23]. We used robust methods to identify variables included in our analyses; for instance, the Cambridge Multimorbidity Score, which outperforms alternatives such as the Charlson Index [34]. Gold standard information on cancer sites was obtained from the NCRAS cancer registry data. Data on patients’ ethnicity—defined in line with UK national census groupings—were identified from the CPRD and HES with 99% completeness. Priority was accorded to ethnicity records in the CPRD over HES, in line with previous recommendations [24,41]. We used combined ethnic categories for simplicity, recognising that this hides some heterogeneity within ethnic subgroups. The alternative to combined grouping was to use the 16 ethnic sub-groups in the 2001 census, which would have reduced power (particularly in rarer cancers) and made the interpretation of our findings unwieldy.

Our cohort was limited to patients with a recorded index feature during the year before diagnosis. This restriction may have introduced bias, as it omits a small number of patients whose first medical contact is in secondary care or emergency departments, and those presenting with non-NICE qualifying symptoms. The demographic characteristics of the excluded patients are broadly similar to those included (results not shown), and as such, the impact of this restriction is likely to be minor, if any.

### 4.2. Comparison with Existing Evidence

The distribution of index features was broadly similar across ethnic groups, although this aspect requires further exploration, given that ethnic minorities may under-report symptoms of possible cancer [14]. The median DI for all participants in our study was 55 days, consistent with recent evidence [30,42], reflecting the downward trend in DIs since the publication of the original NICE guidance on urgent referrals in 2005 [43]. As previously reported [30,43]. DI was shortest for breast cancer, characterised by breast lump, and longest for lung cancer, for which non-specific symptoms such as cough, dyspnoea, and chest infection were frequently recorded. We found evidence of ethnic inequalities in DI across all sites combined, although this requires careful interpretation, considering the strong interaction between ethnicity and cancer sites. As a result, we reported site-specific differences in DI to highlight areas where ethnic inequalities exist.

For colorectal and oesophagogastric cancer, we found longer DI in Asian and Black groups—consistent with previous reports [12,13,18]. Our finding of no difference in DI between White, Asian, and Black patients with ovarian cancer also reaffirms previous reports [12], although this may, in part, reflect a lack of power for the less common cancer site. Likewise, our finding of a longer DI among Black compared with White patients with breast cancer agrees with previous data [12]. In contrast to previous evidence, we found longer DI in Asian and Black patients with prostate cancer and no evidence of differences in Asian patients with breast and lung cancer relative to the White group [12,13,44]. Novel findings include longer DIs in Black patients with myeloma and the Mixed group with lung cancer, in addition to shorter DI in the Other group with myeloma, lung, prostate, and oesophagogastric cancer.

### 4.3. Implications of the Findings

The average DI of 13 days among breast cancer patients in our cohort is encouraging, as this falls well below the 28-day target set in a recent national strategy [45]. We found evidence of ethnic inequality in the DI of breast cancer, although this was small and unlikely to impact patient or clinical outcomes. On average, DI was around two days longer in the Black group than in the White group. Given that over 90% of breast cancers in our cohort had lump as the index feature, with similar proportions across ethnic groups, this finding is unsurprising. However, the average figure masks differences in the 90th centile of DI, which was longer in the Black than the White group (73 days vs 41 days) and may be of significant clinical relevance. While efforts to improve awareness of signs and symptoms (especially non-lump features) and screening uptake are necessary [8,9,10,11,12,13], this finding suggests a need for further exploration of Black women’s pathways to the diagnosis of symptomatic breast cancer.

Our findings of longer DI in Asian and Black patients compared to White patients with prostate cancer may reflect inequalities in primary care. In other words, variations in GPs’ interpretations of reported features—alongside the offer and acceptance of investigation during consultation and prompt referral—may contribute more to the observed ethnic differences at this site. We previously showed that Black men may be less willing to accept prostate-specific antigen (PSA) testing or digital rectal examination, especially at a low perceived risk of prostate cancer [17]. Furthermore, our recent multi-methods study of men with urinary symptoms revealed that GPs are sometimes cautious in offering PSA to Asian and Black men, partly due to the presence of comorbidities [14]. In the present study, recorded index features were similar in White, Asian, and Black men with prostate cancer, although Black men had a slightly higher proportion of morbidity. Even so, this new finding suggests that Asian and Black men with undiagnosed prostate cancer may be receiving differential care, which may adversely impact their outcomes.

We are uncertain why DI was longer in Asian and Black patients with colorectal and oesophagogastric cancer, and in Black patients with myeloma. Until the recent introduction of faecal immunochemical testing to primary care, GPs had to refer all patients with clinical features of suspected colorectal cancer to secondary care for diagnostic investigations [46,47,48]. The same process applies to myeloma and oesophagogastric cancer. GPs’ decisions to refer, the patient’s acceptance of—and attendance for—investigation, and the timeliness of secondary care investigation are important determinants of the DI. It is possible that GPs are applying different referral thresholds or interpreting index features more frequently as benign diseases in Asian and Black patients, thus delaying specialist referral, although we could not study this aspect. Other work suggests that Asian and Black patients with oesophageal cancer were less likely to be diagnosed via the fast-track referral route, although patients from both groups with stomach or colorectal cancer were as likely as White patients to follow this route [25]. The use of the fast-track referral pathway is lower in general practices, with higher concentrations of male and ethnic minority patients (particularly the Asian group) [49]. Both factors applied to Asian and Black patients in our cohort. On the other hand, the observed ethnic differences in the DI of these cancers may arise if Asian and Black patients decline, delay, or are unable to book appointments for specialist investigation, aspects not covered here. Morris et al. found reduced uptake of colonoscopy among UK non-Whites who had a positive Faecal Occult Blood test screening [50]. Further investigation is necessary to unpick the DI of these sites (and others) and determine the main causes of ethnic differences qualitatively.

We found no difference in DI between the Asian, Black, and White groups with lung cancer, in contrast to Neal et al., who reported longer secondary care delays in Asian and Black patients compared to White patients [12]. They reported no evidence of ethnic inequality in total delay, which encompasses patient interval (time from symptom onset to primary care consultation) and DI, a different definition than that used here [12,13].

Our findings of shorter DI among patients of Other ethnic group with myeloma, lung, prostate, or oesophagogastric cancer have not been reported elsewhere, but echoes our recent study showing that patients from this group were more likely than those of other ethnicities to present as emergencies [25]. However, these findings must be interpreted with caution given the heterogeneity within the Other ethnic group (including the Arab, those with unknown or uncategorised ethnicities), with no prior UK studies specifically exploring cancer inequalities in this group.

Overall, the average DI across all sites combined was reasonably short, with modest variations by ethnicity. However, we found site-specific ethnic differences in DI that may concern policymakers and primary care providers. Our finding of a longer DI in Black and Asian men with prostate cancer, although small, cannot be ignored, considering their higher mortality. Our finding of longer DI in Black women with breast cancer, especially among the 10% having DI of over two months, despite the majority presenting with breast lump, may explain their relatively poor outcomes. The deleterious effect of cancer diagnostic delay has now been estimated, with a worse 10-year survival rate of up to 5% for a two-month delay, depending on age. The differences between the White, Asian, and Black groups with myeloma colorectal, or oesophagogastric cancer, are more difficult to explain, but critical to our understanding of ethnic inequalities in cancer outcomes and subject for further inquiries. A 2-month diagnostic delay in oesophagogastric cancer, and of 2–3 months in colorectal cancer, as seen in Asian and Black in our study, may be associated with an estimated 10% reduction in 10-year survival [51].

## 5. Conclusions

Diagnosing cancer in symptomatic patients is especially complicated when the presenting features are non-specific. GPs must balance specialist referral decisions with the need to avoid harm from over-investigation. Patients may be unprepared for—or encounter difficulties in—navigating the diagnostic pathway, leading to delayed diagnosis. Any or a combination of these contextual factors may explain ethnic inequalities in DI among patients diagnosed with five of the seven sites investigated here. While further studies are necessary, this study’s findings enhance our understanding and will help focus interventions to minimise ethnic inequalities in cancer diagnosis.

## Figures and Tables

**Figure 1 cancers-14-03085-f001:**
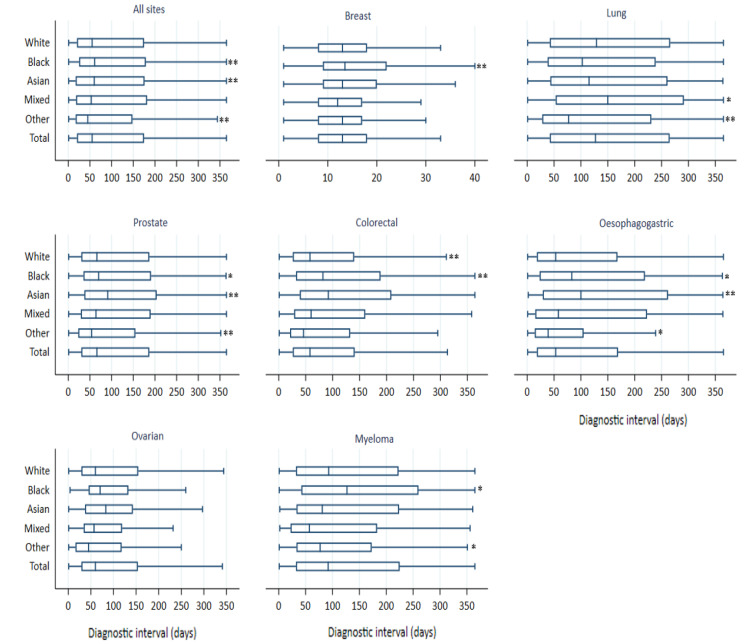
Site-specific box and Whisker plots showing ethnic differences in the number of days to diagnosis (lower adjacent value, 25th, 50th, and 75th centiles, and upper adjacent value). Figures are the exact number of days from index feature to diagnosis. Asterisks indicate significant ethnic differences in DI compared with the White group: (** *p*-value ranges from 0.001 to 0.004) (* *p*-value ranges from 0.01 to 0.05).

**Table 1 cancers-14-03085-t001:** Cancer features sought in participants’ medical records in the year before diagnosis.

Cancer Site	NICE Features
**Breast**	Breast pain, breast lump, breast skin changes (peau d’orange), nipple discharge, nipple retraction, lymphadenopathy (axilla).
**Lung**	Appetite loss, chest infection, chest pain, chest signs consistent with lung cancer, cough, dyspnoea, fatigue, features suggestive of lung metastases’ finger clubbing, haemoptysis, hoarseness, lymphadenopathy (supraclavicular, cervical), shoulder pain, signs of superior vena cava obstruction, stridor, thrombocytosis, weight loss, x-ray findings suggestive of lung cancer.
**Prostate**	Abnormal digital rectal examination, erectile dysfunction, haematuria (visible), nocturia, raised prostate specific antigen (PSA) above age-specific value, urinary frequency, urinary hesitancy, urinary retention, Urinary urgency.
**Colorectal**	Abdominal mass, abdominal pain, change in bowel habit, faecal occult blood, iron-deficiency anaemia, rectal bleeding, rectal mass, weight loss.
**Oesophagogastric**	Back pain, dyspepsia, dysphagia, haematemesis, gastrointestinal bleeding, low haemoglobin, nausea, reflux, suspicious barium meal results, thrombocytosis, upper abdominal mass, upper abdominal pain, vomiting, weight loss.
**Ovary**	Fatigue, abdominal distension/bloating, abdominal or pelvic mass, abdominal pain, abdominal/pelvic mass, appetite loss, ascites, back pain, change in bowel habit, constipation, pelvic pain, raised ca125, urinary urgency, urinary frequency, weight loss.
**Myeloma**	Bone pain, back pain, Bence-jones protein, abnormal erythrocyte sedimentation rate, hypercalcaemia, abnormal white cell count, pathological fracture, plasma viscosity consistent with myeloma, protein electrophoresis suggesting myeloma, spinal cord compression suspected of being caused by myeloma.

**Table 2 cancers-14-03085-t002:** Participant characteristics.

		White	Black	Asian	Mixed	Other	All
**Age**	Median (IQR Years)	72 (63–80)	67 (56–76)	67 (56–75)	70 (62–78)	72 (63–81)	72 (63–80)
**Sex**	Male n (%)	67,263 (57.6)	1768 (70.1)	1175 (54.4)	1352 (58.2)	1650 (55.3)	73,208 (57.8)
**IMD n (%) ***	*1 (least deprived)*	28,413 (24.4)	122 (4.84)	339 (15.7)	453 (19.5)	735 (24.6)	30,062 (23.8)
	*2*	26,240 (22.5)	164 (6.51)	360 (16.7)	424 (18.3)	706 (23.7)	27,894 (22.0)
	*3*	23,353 (19.9)	424 (16.8)	465 (21.5)	471 (20.3)	587 (19.7)	25,200 (19.9)
	*4*	20,029 (17.2)	717 (28.4)	462 (21.4)	502 (21.6)	512 (17.2)	22,222 (17.6)
	*5 (most deprived)*	18,662 (16.0)	1094 (43.4)	533 (24.7)	471 (20.3)	443 (14.9)	21,203 (16.8)
***Morbidity score*** ****** **n (%)**	*0–None*	7814 (6.70)	144 (5.71)	135 (6.25)	115 (4.95)	349 (11.7)	8557 (6.76)
	*1*	17,901 (15.4)	395 (15.7)	299 (13.9)	347 (14.9)	560 (18.8)	19,502 (15.4)
	*2*	26,338 (22.6)	668 (26.5)	503 (23.3)	451 (19.4)	825 (27.6)	28,785 (22.7)
	*3*	29,912 (25.5)	683 (27.1)	641 (29.7)	622 (26.8)	718 (24.1)	32,576 (25.7)
	*4 (most score)*	34,675 (29.7)	632 (25.1)	581 (26.9)	786 (33.9)	533 (17.9)	37,207 (29.4)
**Sites**	Breast	18,280 (15.7)	378 (14.9)	538 (24.9)	441 (19.0)	365 (12.2)	20,002 (15.8)
	Lung	27,926 (23.9)	282 (11.2)	371 (17.2)	460 (19.8)	939 (31.5)	29,978 (23.7)
	Prostate	33,256 (28.5)	1205 (47.8)	572 (26.5)	777 (33.5)	642 (21.5)	36,452 (28.8)
	Colorectal	20,586 (17.7)	311 (12.3)	342 (15.8)	388 (16.7)	524 (17.5)	22,151 (17.5)
	Oesophagogastric	10,102 (8.66)	174 (6.90)	161 (7.46)	135 (5.82)	355 (11.9)	10,927 (8.63)
	Ovarian	3,644 (3.12)	32 (1.27)	94 (4.35)	65 (2.80)	99 (3.32)	3,934 (3.11)
	Myeloma	2,846 (2.44)	140 (5.55)	81 (3.75)	55 (2.37)	61 (2.04)	3,183 (2.51)
	**Total**	**116,640 (92.1)**	**2522 (1.99)**	**2159 (1.71)**	**2321 (1.83)**	**2985 (2.36)**	**126,627 (100)**

IMD: Index of Multiple Deprivation; * Missing record of IMD [n = 46 (0.04%)]; ** quintile group of morbidity score.

**Table 3 cancers-14-03085-t003:** Association between ethnicity and diagnostic interval by cancer site.

Sites	Ethnicity	N	Crude Time Ratio	Adjusted Time Ratio	95% CI	*p*–Value	Average Marginal Difference	95% CI
**All sites**	**White**	**116,640**						
	Black	2522	1.05	1.10	1.05–1.14	<0.001	6.57	3.54–9.60
	Asian	2159	1.02	1.16	1.10–1.22	<0.001	11.0	6.99–15.0
	Mixed	2321	1.00	1.02	0.98–1.07	0.29	1.65	−1.47–4.78
	Other	2985	0.88	0.88	0.84–0.91	<0.001	−8.57	−11.0––6.08
**Breast**	White	18,280						
	Black	378	1.15	1.12	1.04–1.21	0.004	1.55	0.42–2.67
	Asian	538	1.09	1.06	0.99–1.13	0.11	0.72	−0.19–1.62
	Mixed	441	1.00	0.99	0.92–1.06	0.81	−0.11	−0.99–0.77
	Other	365	0.97	0.99	0.92–1.06	0.72	−0.18	−1.11–0.76
**Lung**	White	27,926						
	Black	282	0.90	0.94	0.84–1.04	0.23	−7.26	−18.6–4.23
	Asian	371	0.97	0.98	0.91–1.05	0.51	−2.83	−11.2–5.54
	Mixed	460	1.09	1.06	0.99–1.12	0.08	6.75	−0.85–13.9
	Other	939	0.83	0.90	0.85–0.95	<0.001	−11.5	−17.3––5.62
**Prostate**	White	33,256						
	Black	1205	1.03	1.09	1.03–1.15	0.005	6.63	1.86–11.4
	Asian	572	1.16	1.17	1.09–1.27	<0.001	13.2	6.27–20.0
	Mixed	777	0.99	1.00	0.93–1.08	0.92	0.31	−5.52–6.15
	Other	642	0.87	0.87	0.80–0.95	0.002	−9.68	−15.5–3.88
**Colorectal**	White	20,586						
	Black	311	1.22	1.22	1.09–1.35	<0.001	14.1	5.69–22.4
	Asian	342	1.33	1.37	1.24–1.51	<0.001	24.1	15.5–32.8
	Mixed	388	1.08	1.06	0.96–1.17	0.27	3.85	−3.25–10.9
	Other	524	0.88	0.93	0.85–1.02	0.14	−4.51	−10.3–1.24
**Oesophago–** **gastric**	White	10,102						
	Black	174	1.26	1.21	1.05–1.41	0.01	13.1	2.07–24.1
	Asian	161	1.41	1.33	1.14–1.55	<0.001	19.9	7.51–32.4
	Mixed	135	1.12	1.04	0.86–1.25	0.73	2.11	−9.98–14.2
	Other	355	0.77	0.84	0.74–0.97	0.01	−9.54	−16.5––2.56
**Ovary**	White	3,644						
	Black	32	0.99	0.87	0.66–1.18	0.39	−8.78	−27.4–9.83
	Asian	94	1.09	1.04	0.88–1.24	0.65	2.91	−9.99–15.8
	Mixed	65	0.96	0.96	0.73–1.25	0.73	−3.29	−21.8–15.2
	Other	99	0.83	0.87	0.70–1.09	0.22	−9.13	−22.8–4.59
**Myeloma**	White	2846						
	Black	140	1.15	1.16	1.00–1.35	0.05	15.3	−0.99–31.5
	Asian	81	0.94	0.99	0.82–1.20	0.92	−0.90	−18.9–17.1
	Mixed	55	0.86	0.82	0.62–1.07	0.14	−17.2	−38.1–3.68
	Other	61	0.84	0.83	0.66–1.03	0.09	−16.3	−33.6–0.91

## Data Availability

This study used CPRD-linked data, access to which is subject to protocol approval by an Independent Scientific Advisory Committee, and under which conditions data cannot be shared directly.

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
