# Peer review of "Assessing Ethnic Inequalities in Diagnostic Interval of Common Cancers: A Population-Based UK Cohort Study"

_cancers, 2022, doi:10.3390/cancers14133085_

Round 1

Reviewer 1 Report

Martins et al. use a cohort of cancer patients in the UK to investigate whether there are ethnic inequalities in diagnostic intervals (DI) of seven common cancer types. Given the large size of the cohort, this is a very expansive study of this topic. The authors report ethnic variation in DI for some tumor types and not others. The main novel findings were that Black patients with myelomas and the Mixed group with longer cancer had longer DIs, while much of the other findings confirmed/supported previous findings from other papers. The authors further speculate that this variation in DI may also explain poor outcomes for different ethnic groups. However, I feel this study could be strengthened by addressing the following comments:

Major comment.

1. The authors speculate that ethnic variation in DI may contribute to the poor outcomes in these groups. Is there a way for the authors to provide further data or support for this? Do patients with longer DIs tend to have worse outcomes within an ethnic group? What about between ethnic groups? Or could the longer DI be reflective of a less aggressive tumor presenting weaker symptoms rather than the GP delaying referral? I think providing more evidence for the speculations the authors offer in the discussion or directly linking DI to outcome in this cohort would substantially boost the significance of this paper.

Minor comments

1. typo line 22, 34 (missed space between periods and words). There also seem to be a few other extra or missing spaces throughout.

2. Is a bar (for Total) missing in the Myeloma panel in Figure 1? Can the authors also add asterisks to indicate significantly different comparisons in Fig. 1?

Author Response

Major comment.

Point 1: The authors speculate that ethnic variation in DI may contribute to the poor outcomes in these groups. Is there a way for the authors to provide further data or support for this? Do patients with longer DIs tend to have worse outcomes within an ethnic group? What about between ethnic groups? Or could the longer DI be reflective of a less aggressive tumor presenting weaker symptoms rather than the GP delaying referral? I think providing more evidence for the speculations the authors offer in the discussion or directly linking DI to outcome in this cohort would substantially boost the significance of this paper.

Response 1: The relationship between diagnostic interval and cancer outcomes has been studied extensively, with the commonest view being that it follows a J-shaped curve. The higher mortality associated with shorter diagnostic interval is generally explained by emergency diagnosis or where patient is obviously unwell at presentation, thereby making diagnosis easier. We are not aware of any studies examining the relationship between cancer outcomes and diagnostic intervals by ethnicity. Further, we have no information on cancer biology in our data, and so unable to answer the reviewer’s perfectly valid question on whether longer diagnostic interval seen in Asian and Black patients is reflective of less aggressive tumour presenting. To fully answer the reviewer’s question would require further expansion of the study, which may detract from the main message of the paper. However, we have expanded the discussion section relevant to the reviewer’s query with the following statement

 Our finding of longer DI in Black women with breast cancer, especially among the 10% having DI of over two months, despite the majority presenting with breast lump, may explain their relatively poor outcomes. The deleterious effect of cancer diagnostic delay has now been estimated, with worse 10-year survival of up to 5% for a two-month delay, depending on age”.

 Minor comments

Point 1: typo line 22, 34 (missed space between periods and words). There also seem to be a few other extra or missing spaces throughout.

Response 1: We have now corrected the typos as appropriate, thank you.

Point 2: Is a bar (for Total) missing in the Myeloma panel in Figure 1? Can the authors also add asterisks to indicate significantly different comparisons in Fig. 1?

Response 2: Figure 1 now amended, with asterisks added (and footnotes) as recommended. Thank you.

Reviewer 2 Report

The paper is well written

Some minor suggestions:

Box 1 is not easily readable , please find a improved version to present the cancer features

You reported that "There was strong evidence of interaction between  cancer site and sex (p=0.04).". The effect of gender and its direction should be better explained.

Author Response

Point 1: The paper is well written

Response 1: We appreciate referee’s comment.

Some minor suggestions:

Point 2: Box 1 is not easily readable, please find an improved version to present the cancer features

Response 2: We have now submitted a simpler version of Box 1, thank you.

Point 3: You reported that "There was strong evidence of interaction between cancer site and sex (p=0.04).". The effect of gender and its direction should be better explained.

Response 3: We appreciate the reviewer’s comment and have added the following statement to the results section

There was strong evidence of interaction between site and age, IMD, and morbidity score (p<0.001 for each of these), and between cancer site and sex (p=0.04). The effect of sex was such that women were diagnosed somewhat faster that men in cancers affecting both sexes with the largest difference for oesophagogastric cancer (time ratio=1.17) and smallest difference for lung and colorectal cancer (time ratio=1.10 for both).

Round 2

Reviewer 1 Report

I thank the authors for their responses to the comments and their efforts in revising the manuscript. They have addressed all comments.